# The Development of Fintech and SME Innovation: Empirical Evidence from China

**Hongyu Li [1], Zhiqiang Lu [1,*] and Qili Yin [2]**

1    Business School, Ningbo University, Ningbo 315211, China
2    Audit Department, Liaoning Technical University, Fuxin 123008, China
*    Correspondence: luzhiqiang@nbu.edu.cn; Tel.: +86-574-8760-0383

**Abstract:** Innovation is the source of competitiveness among firms and the driver of economic growth. This paper examines the influence of fintech and firms' innovation from the perspective of stakeholder financial support. To test the relationship, we collect data from Chinese small and medium-sized enterprises (SMEs) between 2011 and 2017. The results indicate that there is a strong positive effect of fintech development on firms' innovative activities. This effect operates through fintech's role in reducing information asymmetry: it increases the availability of funding support from stakeholders for firm R&D investment. Further, the funding supports are not only from investors and banks, but also from other stakeholders. In addition, this effect is larger for bigger and more opaque firms. These findings connect fintech with firm innovation and illuminate the unique roles and mechanisms of fintech development in promoting innovation inputs and outcomes.

**Keywords:** fintech; innovation; R& D investment; information asymmetry; stakeholders

## 1. Introduction

Firm innovation is the source of firm competitiveness and economic growth [1]. However, innovative activities need large and stable funding support [2]. Due to information opacity, irregular financial management, and lack of collateral, small and medium enterprises (SMEs) often face financing constraints for their innovative activities [3]. SMEs are the economic sector making a significant contribution to economic growth and job creation, and inadequate innovation seriously affects their development [4].

In addition, with the extensive development of modern financial technology (fintech) around the world, a number of studies claim that the development of fintech alleviates SME financing constraints by reducing debt costs [5], simplifying credit procedures [6], and promoting inter-bank competition [7]. However, the literature provides little evidence regarding the direct link between fintech and SME innovation. This is a notable gap, given that fintech is already affecting people's lives and firm operations in all areas. This study aims to address this gap by examining the effects of the development of fintech on firm innovation and by unraveling the mechanisms underlying the effect.

The main idea of this paper is that the development of fintech is beneficial for SMEs' access to financing support from stakeholders, which, in turn, is important for SMEs' innovative activities. Stakeholders, including firm investors, creditors, related firms, and regulators, are interested in firm innovation [8], and they are willing to support firms' innovative activities. However, information asymmetry hinders cooperation between firms and stakeholders. In particular, R&D investments are intangible and offer little or no collateral value [9]. As modern Internet technologies, such as big data, cloud computing, and artificial intelligence, have been applied in the financial area, information asymmetry, transaction costs, and cooperation risk between SMEs and stakeholders have decreased [10], making it easier for SMEs to access stakeholders for innovation [11]. Thus, the development of fintech, which is increasing opportunities for SMEs' access to financing from stakeholders, is an important mechanism for SME innovation.

We test these ideas by exploring how fintech development affects SME innovation, and we focus on three main empirical predictions which have not been evaluated previously. We expect the development of fintech to have a positive effect on SME R&D investment; this, in turn, affects innovation outcomes. In addition, we expect that the development of fintech can increase the amount of funding available from stakeholders; this can effectively promote R&D investment by SMEs. Finally, the relationship between fintech development and firm innovation is different for SMEs with different characteristics, as their stakeholders' willingness to participate and their firms' information opacity are different.

Overall, this paper provides evidence that fintech development is beneficial for SMEs in accessing financing from stakeholders; this has a substantial positive impact on SMEs' innovative investment and outcomes. We are aware of no other studies that establish these connections between fintech development, stakeholders' financing, and firm innovation. Given the importance of firm innovation for the development of productivity and for the economy, this paper establishes a previously unexplored channel that connects fintech development and economic growth. The remainder of this paper proceeds as follows. Section 2 proposes a theoretical analysis framework. Section 3 describes sample construction, data sources, variables, and empirical models. Section 4 presents the main empirical analysis and robustness tests. Section 5 is the implications and discussions. Finally, Section 6 concludes this study.

## 2. Theoretical Analysis

Firms need a large and stable cash flow to support technological innovation; this is often characterized by high risk, a long cycle, and high investment, and there is not always sufficient capital to perform R&D investment [12]. Firms need the financial support of stakeholders—for example, equity investment by outside shareholders [13], loans from banks [14], trade credit from upstream firms [15], and subsidies from the government [16]. However, the information asymmetry problem is widespread between firms and their stakeholders [17], and SMEs often cannot provide enough "hard" information to their stakeholders; this exacerbates the information asymmetry problem [18]. Specifically, information asymmetry leads to poor information transmission, aggravated enterprise risks, and inefficient resource collection, which limits firms' financing channels and hinders firms' financing from stakeholders.

With the development of fintech, these problems might be solved. The term "fintech" is a neologism that originates from the words "financial" and "technology" [19]. It describes, in general, the connection of Internet-related technologies (e.g., big data, cloud computing) with established business activities in the financial services industry [20]. Through modern Internet technology, fintech can improve information transmission, risk management, and resource allocation in modern finance [21], and a fintech ecosystem can be constructed by the interaction of firms and their stakeholders [22]. Fintech facilitates information transmission [23]. SMEs can trade through digital platforms and can create a historical record. For example, they can open digital accounts and use digital payment channels to pay or receive money [24], or they can operate on e-commerce platforms and adopt digital invoices [25]. Digital platforms improve the efficiency of trade and payment settlement. Meanwhile, digital history enriches the existing information of SMEs [26], and it can be trusted by third parties because digital data cannot be changed [27].

In addition, fintech improves risk management capabilities. Through fintech technologies such as big data, cloud computing, and artificial intelligence, stakeholders can use nontraditional finance data such as mobile phone and digital account usage to develop alternative credit scores, which make it possible to extend credit to SMEs without detailed financial data or transaction records [28]. Therefore, the development of fintech widens information sources and reduces the risks of cooperation between stakeholders and SMEs.

Finally, fintech can effectively enhance the efficiency of resource allocation. Fintech is characterized by modern Internet-related technology such as big data, cloud computing, and artificial intelligence. The speed of automated algorithms is far faster than human

calculations [29], and this reduces the transaction cost, improves the efficiency of risk management, and expands the boundaries of transaction possibilities [30]. Internet-related technologies make it possible for the supply and demand of money to trade directly [31], such as with P2P or online lending. In addition, firm stakeholders are more likely to supply capital directly to SMEs' innovation activities, rather than through financial intermediaries such as banks.

In sum, the development of fintech can improve information transmission, risk management, and resource allocation in modern finance, which can improve financing channels between SMEs and their stakeholders. Therefore, stakeholders are more likely to cooperate with SMEs, and support them with capital. This, in turn, can increase SMEs' R&D investment and their ability to pursue innovative projects. Therefore, the relationship between the development of fintech and firm innovation is visually illustrated in Figure 1.

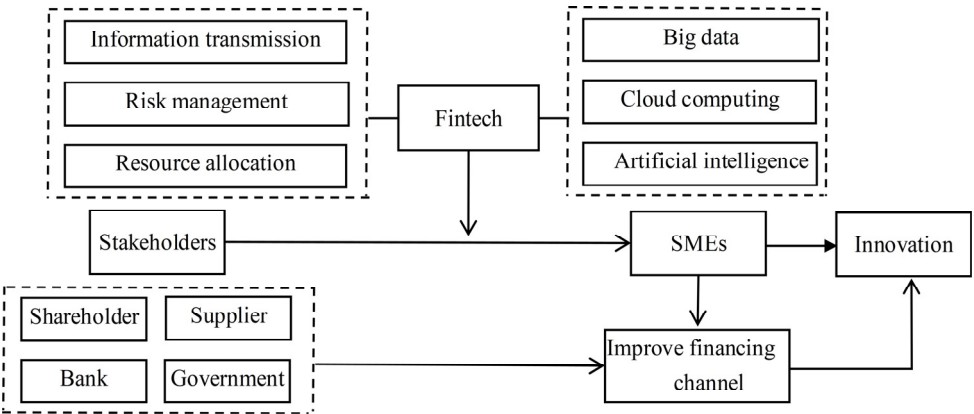

**Figure 1.** The relationship between the development of fintech and firm innovation.

Based on theoretical analysis and Figure 1, we expect that the development of fintech positively affects firms' innovative outcomes.

## 3. Methodology

### 3.1. Data

Our data came from several different databases. Firm patents, R&D investment, and financial information were collected from the China Stock Market & Accounting Research (CSMAR) database. The macroeconomic information was collected from the China Statistical Yearbook issued by the National Bureau of Statistics of China.

The initial firm sample included all listed companies in the SME Board and the growth enterprise market (GEM) from 2011 to 2017. To complete our study, we processed the data as follows: (1) we excluded the observations with missing values in key variables; (2) financial and utility firms were excluded to make the observations' information more comparable; (3) we excluded observations from the Tibet province because there are some abnormal and missing values in the Tibet macroeconomy. The final sample consists of 7169 firm-year observations from 1128 SMEs.

### 3.2. Variables

#### 3.2.1. Innovation

Following the existing literature [32–35], we measured corporate innovation by patent metrics. The Patent Database in CSMAR provides detailed information on patents applied for by listed companies and granted by the State Intellectual Property Office (SIPO) of China from 2007 to 2017. Based on the information available in the database, we constructed two measures for firm years' innovation outcomes. The first measure was the company's number of patent applications (APPLY), and the second measure was the company's number of patents that were eventually granted by the SIPO (GRANT). In the regression model, we used the natural logarithm of the total number of patent applications and

the natural logarithm of the patents generated (logAPPLY and logGRANT) as the main innovation measures in our analysis. Meanwhile, to avoid losing observations with zero patents, we added one to the actual values when calculating the natural logarithm.

In addition, following the relevant studies [36,37], we also considered SME innovation input. The measure of innovation input was the R&D investment of firms (R&D). In the regression model, we also used the natural logarithm of R&D investment of firms (logR&D) and added one to the actual values when calculating the natural logarithm.

### 3.2.2. The Development of Fintech

Following the relevant literature [21,38,39], we used the 'Baidu Index' in Baidu, the largest search engine in China, to construct the variable of fintech. Considering that there are four main functions of modern finance, including information transmission, risk management, resource allocation, and payment settlement [40], and noting the technological foundation of fintech (such as cloud computing, big data, blockchain, and so on), we constructed fintech key words from five dimensions (Table 1). We then counted the number of daily news searches that contained each word in Table 1 in each province during the sample period.

**Table 1.** Fintech word list.

| Dimension | Keywords | | | |
|---|---|---|---|---|
| Information transfer | E-bank | Internet bank | Online bank | Abbreviation for online bank | Network banking |
| Risk management | Internet finance | Credit information system | Online insurance | Online finance | Internet insurance |
| Resource allocation | Online lending | P2P | Internet investment | Crowdfunding | Smart investment |
| payment | Cross-bank clearing | Online payment | Internet payment | Mobile payment | Third-party payment |
| Technical base | Big data | Cloud computing | Artificial intelligence | Blockchain | Strategic decision support |

The attention to the relevant news represents the development of a certain thing in society [41]. Therefore, the more attention paid to the news related to the keywords in Table 1, the more attention and opinions received from the public, indicating a more prosperous development of fintech [21]. We counted the number of daily news searches that corresponded to each keyword listed in Table 1 from each province, and we added them up to calculate the total number of all of the keywords in each province during the sample period as the variable of fintech development (Fintech). In the regression model, we used the natural logarithm of the number of daily news searches in each province, and we added one to the actual values when calculating the natural logarithm (FINTECH).

### 3.2.3. Control Variables

In our regression analysis, we used a set of firm-level and province-level control variables. Following the relevant literature [42], we included the size of the firm (Log SALES), which is the logarithm of lagged sales. Our specification also included asset tangibility (PPE), which was the net value of the property plant and equipment divided by total assets. In addition, we also controlled for firm return on assets (ROA), firm leverage (LEV), firm age (AGE), and the proportion of the biggest shareholder (OWN1).

We also controlled for some province-level macroeconomic variables by referring to [43]. We controlled for the natural logarithm of the provincial GDP per capita (log GDPC), the degree of inflation in each province (CPI), and the degree of government intervention using the proportion of the government's financial expenditure on regional GDP (GOVERN). In addition, we computed the province-level labor force composition

for 19 different industry segments following formal industry classification, and we controlled for province-level labor force concentration using the Herfindahl Index (LOB-CON). Meanwhile, we also computed the province-level industrial share of total value added, and we controlled for province-level industrial concentration using the Herfindahl Index (VALCON).

Table 2 provides a statistical summary of our main variables. In terms of SME innovation, the average number of patent applications was 38, of which 27 were granted. The average R&D investment was 58 million, and there was great variation among firms. In terms of the development of fintech, the average number of daily news searches was 4302, and there was a great difference among the provinces. The fewest provinces had only 165 daily news searches, while the most provinces had about 9364.

**Table 2.** Descriptive statistics.

| Variables | Obs | Mean | Median | SD | Min | Max |
|---|---|---|---|---|---|---|
| APPLY | 7169 | 37.514 | 17.000 | 61.881 | 0.000 | 395.000 |
| GRANT | 7169 | 27.082 | 11.000 | 46.426 | 0.000 | 301.000 |
| R&D (million) | 7169 | 58.91 | 37.21 | 60.33 | 0.000 | 222.54 |
| Fintech | 7176 | 4302 | 3844 | 2244 | 165 | 9364 |
| SALES (billion) | 7169 | 1.54 | 9.09 | 1.62 | 0.184 | 6.22 |
| ROA | 7169 | 0.051 | 0.048 | 0.036 | −0.007 | 0.132 |
| PPE | 7169 | 0.194 | 0.173 | 0.124 | 0.022 | 0.463 |
| LEV | 7169 | 0.324 | 0.301 | 0.179 | 0.068 | 0.671 |
| AGE | 7169 | 12.932 | 13.000 | 4.186 | 5.000 | 21.000 |
| OWN1 | 7131 | 0.343 | 0.329 | 0.133 | 0.147 | 0.612 |
| GDPC | 7169 | 62,455 | 63,374 | 20,872 | 24,446 | 99,995 |
| CPI | 7169 | 2.693 | 2.303 | 1.204 | 0.567 | 6.338 |
| GOVERN | 7169 | 0.166 | 0.148 | 0.045 | 0.110 | 0.250 |
| LABCON | 7169 | 0.180 | 0.197 | 0.061 | 0.076 | 0.286 |
| GDPCON | 7169 | 0.237 | 0.236 | 0.020 | 0.194 | 0.277 |

*3.3. Empirical Model*

To assess the way in which the development of fintech affected SME innovation, we estimated the following model:

$$Innovation = \alpha + \beta_1 FINTECH_{jt-1} + \gamma Z_{ijt-1} + Industy_k + Year_t + \varepsilon_{it} \qquad (1)$$

where $i$ indexes firm, $t$ indexes time, $j$ indexes province, and $k$ indexes industry. The dependent variables in Equation (1) are the variables of innovation, including the natural logarithm of the number of patents applied (logAPPLY), the natural logarithm of the number of patents granted (logGRANT), and the natural logarithm of R&D investment (logR&D) in a firm. The core independent variable is the development of fintech (FINTECH). $Z$ is a vector of control variables that includes firm-level and province-level variables. $\alpha$ is the intercept term of the regression model, $\beta_1$ is the regression coefficient of the variable FINTECH, and $\gamma$ is the regression coefficient of the control variable. Based on the analysis in the theoretical reasoning, we expected that the development of fintech could increase firm innovation, i.e., $\beta_1 > 0$. $Industry_k$ and $Year_t$ capture industry and year fixed effects, respectively. To avoid the endogenous problem caused by potential reverse causality, control variables were lagged by one period when included in the model.

In addition, to analyze the influence mechanism of fintech development on firm innovation, we estimated the impact of fintech development on stakeholders' funding supports. The regression model was as follows:

$$Stakeholders'\ funding = \alpha + \beta_1 FINTECH_{jt-1} + \gamma Z_{ijt-1} + Industy_k + Year_t + \varepsilon_{it} \qquad (2)$$

The dependent variables in Equation (2) are the variables of stakeholders' funding supports to firms, including shareholders, banks, upstream firms, and the government. The

proxy variable of shareholders' funding supports is the absorbed shareholder investment divided by the total assets (Equity). Banks' financial support is access to bank loans divided by total assets (Loan), upstream firms' financial support is trade credit divided by total assets (Payable), and the government's financial support is tax subsidies divided by total assets (Tax). The core independent variable and the control variables in Equation (2) are the same as in Equation (1).

## 4. Results

### 4.1. Baseline Results

Table 3 presents the results of the baseline estimations. In terms of firm-level control variables, company size is positively correlated with innovation, but fixed assets ratio and ownership concentration is negatively correlated with innovation. Turning to the province-level control variables, the findings suggest that firm innovation is positively associated with the government's financial expenditure, labor concentration, and industrial concentration.

**Table 3.** Baseline regression results.

| Dependent Variable | (1) logAPPLY | (2) logGRANT | (3) logR&D | (4) logAPPLY | (5) logGRANT |
|---|---|---|---|---|---|
| FINTECH | 0.435 | 0.400 | 0.606 | | |
| | (0.056) *** | (0.055) *** | (0.178) *** | | |
| log R&D | | | | 0.061 | 0.052 |
| | | | | (0.003) *** | (0.003) *** |
| log SALES | 0.397 | 0.357 | 0.572 | 0.363 | 0.332 |
| | (0.021) *** | (0.021) *** | (0.066) *** | (0.019) *** | (0.019) *** |
| ROA | 1.054 | 0.591 | 1.333 | 0.970 | 0.571 |
| | (0.484) ** | (0.481) | (1.543) | (0.442) ** | (0.440) |
| PPE | −1.113 | −1.081 | −1.504 | −1.121 | −1.094 |
| | (0.136) *** | (0.135) *** | (0.434) *** | (0.124) *** | (0.124) *** |
| LEV | −0.045 | 0.135 | −1.918 | −0.015 | 0.126 |
| | (0.116) | (0.115) | (0.369) *** | (0.106) | (0.106) |
| AGE | 0.002 | −0.004 | −0.039 | 0.002 | −0.003 |
| | (0.004) | (0.004) | (0.013) *** | (0.004) | (0.004) |
| OWN1 | −0.272 | −0.173 | −1.079 | −0.295 | −0.205 |
| | (0.117) ** | (0.116) | (0.371) *** | (0.107) *** | (0.106) * |
| log GDPC | −0.222 | −0.285 | 0.287 | −0.008 | −0.093 |
| | (0.062) *** | (0.062) *** | (0.199) | (0.049) | (0.049) * |
| CPI | −0.029 | −0.041 | −0.037 | 0.002 | −0.002 |
| | (0.045) | (0.045) | (0.143) | (0.033) | (0.033) |
| GOVERN | 1.660 | 0.902 | 0.530 | 1.050 | 0.323 |
| | (0.602) *** | (0.597) | (1.916) | (0.557) * | (0.554) |
| LABCON | 1.496 | 1.783 | −0.211 | 2.047 | 2.273 |
| | (0.398) *** | (0.394) *** | (1.266) | (0.353) *** | (0.352) *** |
| GDPCON | 2.854 | 2.710 | 6.571 | 5.349 | 4.987 |
| | (1.067) *** | (1.059) ** | (3.399) * | (0.906) *** | (0.902) *** |
| Sector dummies | Yes | Yes | Yes | Yes | Yes |
| Year dummies | Yes | Yes | Yes | Yes | Yes |
| N | 7131 | 7131 | 7131 | 7131 | 7131 |
| R2 | 0.263 | 0.244 | 0.314 | 0.309 | 0.275 |

Note: Standard errors in parentheses; * $p < 0.10$, ** $p < 0.05$, *** $p < 0.01$.

Regarding the key variable in the paper, in Columns (1) and (2), the coefficients of FINTECH are positive, and they are statistically significant at the 1% level. This finding supports the idea that the development of fintech leads to an increase in the number of firm patents, both in patent applications and patent grants. In addition, we find that the positive relationship between fintech and firm innovation is also important in economic terms. According to the coefficient estimates of FINTECH in Columns (1) and (2), a 1% increase in

fintech development results in 0.435% and 0.4% more patent applications and patent grants to the firm in the region.

Further, we analyze the impact of fintech development on firm R&D investment. In Column (3), the coefficient of FINTECH is positive, and it is statistically significant at the 1% level. In addition, the coefficient of FINTECH shows that a 1% increase in fintech development will lead to a 0.606% increase in R&D investment. The finding indicates that the development of fintech can increase a firm's R&D investment, which might lead to an increase in firm patents. The results in Columns (4) and (5) support this assertion, since an increase in R&D investment leads to an increase in patents, both in patent applications and in patent grants.

### 4.2. Mechanism Tests

Our evidence so far shows the positive effects of fintech development on SME innovation. In this section, we explore the way in which fintech development affects SME innovation. Based on theoretical reasoning, it is presumed that fintech development can effectively reduce information asymmetry between a firm and its stakeholders. As a result of building trust between them, there can be an increase in funding support by stakeholders of the firm, and then a boost in the firm's R&D investment. There are different forms of funding support by firm stakeholders; among them are times when investors provide equity input, when banks provide bank loans, when suppliers provide trade, and when the government provides tax subsidies. Therefore, we analyze the intermediate mechanisms from the perspectives of investor input, bank loans, trade credit, and tax subsidies.

We report the estimated results in Table 4. Panel A tests the effect of fintech development on stakeholders' funding support, and panel B tests the impact of stakeholders' funding supports on firm R&D investment. The estimation results in Panel A of Table 4 show that the coefficients of FINTECH are significantly positive; this suggests that the development of fintech increases stakeholders' funding supports to the firm.

Furthermore, the estimated results in Panel B of Table 4 show that the coefficients of Loan, Payable, and Tax are positive and significant at a 5% and 1% level, respectively. Although the indicator that represents investor input is not significant, the coefficient is also positive. This finding indicates that stakeholders' providing of money to the firm would increase the R&D investment of SMEs, so that the innovation outcomes of SMEs may be increased.

Based on the results of this analysis, stakeholders' funding support appears to be an essential intermediate mechanism through which fintech development affects firm R&D investment and innovation outcomes.

### 4.3. Robustness Checks

In this section, we check the robustness of the baseline findings. We first examine whether our results are robust to alternative proxies for innovation, fintech development, and control variables. In addition, we use the two-stage least square (2SLS) method to eliminate the endogenous problem in the model. Finally, we summarize the regional SME innovation data to analyze the impact of regional fintech development on the total number of innovations in the provinces.

#### 4.3.1. Alternative Proxies for Innovation, Fintech Development, and Control Variables

We use the patents of listed companies to measure SME innovation, and Columns (1) and (2) of Table 5 report the estimation results. The coefficient estimation of FINTECH is positive and statistically significant at a 1% level, which means the results are basically unaltered.

In addition, according to the study of [44], we use the number of fintech companies (in the form of the natural logarithm) registered in a year in a province to measure the regional development of fintech, and the results are shown in Columns (3) and (4). The coefficients of FINTECH companies are significantly positive with firm innovation, which is in line with our baseline findings.

**Table 4.** Mechanism regression results.

| Panel A: The dependent variable is fund supply of stakeholders | | | | |
|---|---|---|---|---|
| Dependent Variable | (1)<br>Equity | (2)<br>Loan | (3)<br>Payable | (4)<br>Tax |
| FINTECH | 0.012<br>(0.005) *** | 0.010<br>(0.006) * | 0.004<br>(0.002) ** | 0.002<br>(0.001) *** |
| Control variables | Yes | Yes | Yes | Yes |
| Sector dummies | Yes | Yes | Yes | Yes |
| Year dummies | Yes | Yes | Yes | Yes |
| N | 6362 | 6794 | 7127 | 6696 |
| $R^2$ | 0.158 | 0.440 | 0.276 | 0.066 |
| **Panel B: The dependent variable is firm R&D investment** | | | | |
| Dependent Variable | (1)<br>logR&D | (2)<br>logR&D | (3)<br>logR&D | (4)<br>logR&D |
| equity | 0.785<br>(0.523) | | | |
| loan | | 0.993<br>(0.452) ** | | |
| payable | | | 7.479<br>(1.066) *** | |
| tax | | | | 28.008<br>(3.509) *** |
| Control variables | Yes | Yes | Yes | Yes |
| Sector dummies | Yes | Yes | Yes | Yes |
| Year dummies | Yes | Yes | Yes | Yes |
| N | 7374 | 7804 | 8139 | 7692 |
| $R^2$ | 0.358 | 0.361 | 0.366 | 0.364 |

Note: Control variables are the same as in Table 3. Standard errors in parentheses; * $p < 0.10$, ** $p < 0.05$, *** $p < 0.01$.

**Table 5.** Alternative measure of variables.

| | Innovation = Patents for Invention | | Digital Finance = the Number of Fintech Companies | | Aggregation of Firm-Level Controls | |
|---|---|---|---|---|---|---|
| | (1) | (2) | (3) | (4) | (5) | (6) |
| | logAPPLY | logGRANT | logAPPLY | logGRANT | logAPPLY | logGRANT |
| FINTECH | 0.341<br>(0.056) *** | 0.291<br>(0.053) *** | | | 0.446<br>(0.060) *** | 0.408<br>(0.060) *** |
| COMPANIES | | | 0.121<br>(0.023) *** | 0.116<br>(0.023) *** | | |
| Control variables | Yes | Yes | Yes | Yes | Yes | Yes |
| Sector dummies | Yes | Yes | Yes | Yes | Yes | Yes |
| Year dummies | Yes | Yes | Yes | Yes | Yes | Yes |
| N | 7131 | 7131 | 7131 | 7131 | 7169 | 7169 |
| R2 | 0.193 | 0.205 | 0.260 | 0.241 | 0.204 | 0.190 |

Note: Control variables are the same as in Table 3. Standard errors in parentheses; *** $p < 0.01$.

Further, we aggregate the firm-level controls at the province–industry–year level, as is typically reported in the literature [45]. We replace the firm-level variables with the average values for the size, performance, fixed asset ratio, share concentration, and age of firms belonging to the same industry in the given year; the results are shown in Columns (5) and (6) of Table 5. The coefficients of FINTECH are also positive and are statistically significant at the 1% level, which means that the regression results are unaltered.

4.3.2. Two-Stage Least Square (2SLS)

In this sub-section, we replace the OLS model in the baseline regression with the two-stage least square (2SLS) method to solve the endogenous problem caused by any omitted variables. In the model, we use "the distance between the cities where the firm is located and Hangzhou which is the birthplace of fintech in China" (DISTANCE) as the instrumental variable [46]. Although the main realization form of fintech is online, the development degree is still affected by geographical spatial factors, and the farther away from Hangzhou, the more difficult it is to promote. Therefore, "the distance between the cities where the firm is located and Hangzhou" is directly related to the development of fintech but is not directly related to firm innovation. In addition, distance is an exogenous variable that cannot change with the development of the regional society and economy.

Column (1) of Table 6 is the first stage regression result, which shows that "the distance between the cities where the firm is located and Hangzhou" is significantly negatively correlated with the development of fintech. The result confirms the assertion that the development degree of fintech is affected by geographical spatial factors. Columns (2), (3), and (4) of Table 6 are the second-stage regression results; the coefficients of FINTECH are also significantly positive, indicating that the results of baseline regression are robust.

**Table 6.** Two-stage least square regression results.

| Dependent Variable | (1) | (2) | (3) | (4) |
| :---: | :---: | :---: | :---: | :---: |
| | **FINTECH** | **logAPPLY** | **logGRANT** | **logR&D** |
| DISTANCE | −0.026 | | | |
| | (0.001) *** | | | |
| FINTECH | | 0.256 | 0.195 | 0.976 |
| | | (0.092) *** | (0.091) ** | (0.259) *** |
| Control variables | Yes | Yes | Yes | Yes |
| Sector dummies | Yes | Yes | Yes | Yes |
| Year dummies | Yes | Yes | Yes | Yes |
| N | 7131 | 5998 | 5998 | 5998 |
| R2 | 0.834 | 0.260 | 0.244 | 0.340 |

Note: Control variables are the same as in Table 3. Standard errors in parentheses; ** $p < 0.05$, *** $p < 0.01$.

4.3.3. Provincial Panel Data Analysis

In the baseline regression, we use firm-level samples to analyze the relationship between fintech and firm innovation. In this sub-section, we check whether our results are robust when we use provincial panel data. We aggregate the number of patent applications, patent grants, and R&D spending in our samples at the province–year level. The results of the provincial panel data analysis are shown in Table 7.

In Table 7, we find that the coefficient estimates of FINTECH are all positive and significant for patent applications, patent grants, and firm R&D investment at the 1% level. The finding supports the conclusion that fintech leads to an increase in firm innovation input and outcomes; this is consistent with our baseline analysis.

*4.4. Heterogeneity Analysis*

Although the results, thus far, indicate that fintech development can boost SME innovation, there might be heterogeneity caused by firms' specific characteristics. For example, the firm size would influence the willingness of stakeholders to cooperate [47], and the firm opacity would affect the degree of information asymmetry between SMEs and their stakeholders [48]. As such, in this section, we analyze heterogeneity from two perspectives: firm size and firm opacity.

**Table 7.** Provincial panel data analysis.

| Dependent Variable | (2) | (3) | (4) |
| :---: | :---: | :---: | :---: |
| | **logAPPLY** | **logGRANT** | **logR&D** |
| FINTECH | 2.497 | 2.493 | 2.229 |
| | (0.020) *** | (0.021) *** | (0.016) *** |
| Province control variables | Yes | Yes | Yes |
| Province dummies | Yes | Yes | Yes |
| Year dummies | Yes | Yes | Yes |
| N | 7169 | 7169 | 7169 |
| R2 | 0.841 | 0.837 | 0.873 |

Note: Province control variables including logGDPC, CPI, GOVERN, LABCON, and GDPCON. Standard errors are in parentheses. *** $p < 0.01$.

### 4.4.1. Firm Size

To analyze the impact of firm size on the relationship between fintech and innovation, we construct a dummy variable, SMALL, which equals 1 for firm years below the industry median total asset and 0 for firm years above the industry median total asset.

We report the results in Columns (1) and (2) of Table 8. The coefficient estimates of FINTECH are also positive and significant at the 5% level. The coefficient estimates of the interaction term (FINTECH × SMALL) are negative in both specifications and are significant at a 1% level. Based on the coefficient estimate of the interaction terms reported in Columns (1) and (2), the marginal effects of fintech for bigger firms are 0.490 and 0.451, but the marginal effects are only 0.343 (0.490–0.147) and 0.316 (0.451–0.135), which means that, as the fintech development grows by 1%, patent applications and patent grants increase by more than 0.147% and 0.135% for the bigger firms.

**Table 8.** Heterogeneity analysis.

| Dependent Variable | (1) | (2) | (3) | (4) |
| :---: | :---: | :---: | :---: | :---: |
| | **logAPPLY** | **logGRANT** | **logAPPLY** | **logGRANT** |
| FINTECH | 0.490 | 0.451 | 0.421 | 0.389 |
| | (0.060) *** | (0.060) *** | (0.056) *** | (0.055) *** |
| SMALL | 0.959 | 0.900 | | |
| | (0.430) ** | (0.427) ** | | |
| FINTECH × SMALL | −0.147 | −0.135 | | |
| | (0.052) *** | (0.052) *** | | |
| OPACITY | | | 0.040 | 0.091 |
| | | | (0.372) | (0.369) |
| FINTECH × OPACITY | | | 0.019 | 0.015 |
| | | | (0.004) *** | (0.004) *** |
| Control variables | Yes | Yes | Yes | Yes |
| Sector dummies | Yes | Yes | Yes | Yes |
| Year dummies | Yes | Yes | Yes | Yes |
| N | 7131 | 7131 | 7129 | 7129 |
| R2 | 0.225 | 0.210 | 0.226 | 0.224 |

Note: Control variables are the same as in Table 3. Standard errors in parentheses; ** $p < 0.05$, *** $p < 0.01$.

These findings suggest that fintech development enhances the innovation outcomes for bigger SMEs, since the stakeholders are more willing to cooperate with firms within a certain size.

### 4.4.2. Firm Opacity

Following the method of the relevant literature [49], we set the dummy variable of OPACITY, which equals 1 for firm years above the industry median intangible assets ratio

(i.e., more opaque) and 0 for firm years below the industry median intangible assets ratio (i.e., less opaque).

We report the results in Columns (3) and (4) of Table 8. The coefficient estimates of FINTECH are positive in both specifications and are significant at the 1% level, which suggests that fintech development has a positive effect on innovation outcomes by firms with lower opacity. Further, the coefficient estimates of the interaction term (FINTECH × OPACITY) are positive in both specifications and are significant at the 1% level, which means that firms with higher opacity are more affected by fintech development in terms of innovation. Based on the coefficient estimate of the interaction terms reported in Columns (3) and (4), fintech development grows by 1%, and patent applications and patent grants increase by 0.019% and 0.015%, respectively, for firms with higher opacity.

Because firms with higher opacity have a higher degree of information asymmetry with their stakeholders, stakeholders are less willing to cooperate with them. In this case, fintech will play a greater role in reducing information asymmetry and in promoting cooperation between SMEs and their stakeholders; this enhances the relationship between fintech and firm innovation.

## 5. Discussions

### 5.1. Implications

Given the increasing importance of innovation and fintech, this paper can have important academic and policy implications. First, this paper contributes to the literature on the real role of fintech by providing insights into the consequences fintech development has on firm innovation. Consistent with the studies [50–52], the results of this paper illustrate that financing constraint matters for SME R&D investment. Further, this paper provides evidence on the way in which the role of fintech development may mitigate this problem and serve to promote a firm's innovative activities; this is an important channel through which fintech can boost the development of economic growth.

Second, existing research indicates that information asymmetry is an important reason for SME financing constraints in innovative activities [53–55]. This paper analyzes the role of fintech in alleviating information asymmetry between SMEs and their stakeholders. Different from existing research on financing constraints [56], this paper suggests that financing supports for SMEs' innovative activities should include not only investors and banks, but also upstream firms and the government.

Finally, although this study uses only Chinese data, other countries (especially other emerging or developing countries) can also learn from this research. In order to minimize SME financing constraints in innovative activities and to promote innovative activities in firms in the process of financial development, the governments of countries with underdeveloped capital markets should further encourage the combination of modern Internet technology (such as big data and cloud computing) and finance. At the same time, with the revolution in the area of finance by fintech development, special regulations, policies, and approaches should be applied to decrease financial risk created by fintech development and to promote the healthy development of financial markets.

### 5.2. Limitations and Implementations

Despite having several contributions, this research has limitations that should be taken into account in future studies. First, the analysis in this paper is limited to SMEs only. However, the firm size may be an important factor in the relationship between fintech and innovation. To explore the influence of fintech on different types of firms, there is much to be done to expand the number of samples and further examine this area. Second, we use only five dimensions to construct a fintech word list and measure the development of fintech. Other dimensions (such as finance digital transformation, central bank digital money, intelligent investment manager, etc.) should be considered in future studies. Finally, we analyze the relationship between the development of fintech and firm innovation, but we do not further investigate its influencing mechanism. Future studies

could further explore other factors influencing this relationship, such as macroeconomic policies, environmental uncertainty, corporate governance, etc.

## 6. Conclusions

Innovation is a key factor in a firm's competitiveness and economic growth, and the emergence of fintech is a great revolution in the modern financial industry. Yet, few studies examine whether and how fintech development impacts firm innovation, especially in emerging or developing countries. This paper tries to systematically explore the role of fintech development in firm innovation and unravel the mechanisms through which this effect occurs. Using 1128 Chinese SMEs from 2011 to 2017, we provide strong evidence that fintech development increases firm innovative inputs and outcomes. Further, this paper suggests that fintech development is beneficial for SMEs' ability to cooperate with their stakeholders and to gain access to financing support from stakeholders; this can effectively increase a firm's R&D investment. Finally, this paper finds that the positive effect of fintech development is significantly larger for firms of a larger size, indicating that stakeholders are more willing to cooperate with firms of a certain size.

**Author Contributions:** Conceptualization, H.L.; methodology, Z.L.; software, Z.L. and Q.Y.; formal analysis, H.L.; investigation, Q.Y.; resources, Q.Y.; data collection, Q.Y.; writing—original draft preparation, H.L.; writing—review and editing, Z.L.; supervision, H.L.; funding acquisition, H.L. and Z.L. All authors have read and agreed to the published version of the manuscript.

**Funding:** This research was funded by the Natural Science Fund of Zhejiang Province, grant number LY22G030010, and the National Social Science Fund of China, grant number 20BJY158.

**Conflicts of Interest:** The authors declare no conflict of interest.

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
