# Peer review of "The Development of Fintech and SME Innovation: Empirical Evidence from China"

_sustainability, doi:10.3390/su15032541_

Round 1
Reviewer 1 Report
Dear Authors,
It was a great pleasure to read your research. Congratulations.
It is a very interesting topic and makes a very positive contribution to science and management.
Kindest regards
Note: on line 289, please correct Table 4, Panel B (instead of Table 4B).
Reviewer 2 Report
I congratulate you on this interesting topic and well written research paper.
My major concerns are the stability of the empirical results, ie. robustness analyses regarding the word lists as well as different robustness tests.
Furthermore please check the grammar and spelling as I detected several typos.
good luck!
Reviewer 3 Report
The article presents a very interesting study that can be used as an example by other countries, as well as a prelude to further in-depth research in the development of innovation in SMEs.
The aims and objectives of the article are clearly stated, as are the assumptions for the research. The results of the research are convincing. I have only a few minor comments on the introduction, the discussion of the research results and the conclusion:
- in the introduction the authors too often use the phrases: our study, our results, our findings. In a scientific article, the impersonal form can be used.
- the discussion lacked a broader reference to other studies conducted in the area under analysis
- in the conclusion, the Authors could point out the limitations of the conducted research.
Reviewer 4 Report
The paper addresses an interesting topic of research. Innovation in an important characteristic that each company mut have so they can adapt to the new and more demanding challenges in the socio-economic environment.
The introduction part offers some information regarding the context of the research but I think that it can be improve a little bit. I consider that you present to much of your findings in the introduction part. In this section of the paper, I expect to see an explanation of the context of the research, the research questions, basically this section will familiarize the reader with the topic of your research. I consider that you can keep all the information presented here and move it in the results or conclusions section and in the introduction just highlight the main research gaps and how do you approach those gaps.
The next section of theoretic analysis you presented well the meaning of the fintech term but I did not understand if figure 1 is a figure proposed by you. Judging by the lack of citation I supposed you proposed that figure, if this is the case you need to explain how you generate it and maybe you need to move it in the results section pf the paper. Until this point, in the introduction and literature background you do not need to present your contribution, they are sections where the readers are getting familiar with the subject. All the contributions must be presented after you present the methodology.
I consider that you need to avoid bulk citation and for each paper that helped you in the research you can write some words and indicate exactly their help.
When you use formulas you can insert the explanation for each terms under the formula, in your case I did not find any explanation about what α or β is and so on. You need just to add the explanation of the terms.
I also suggest not to use expression as “first”, “second” etc. you can express your ideas directly.
Also being a subject of innovation I consider that you can take into consideration more actual papers for your study. Your first citation is from 1992 and on you entire reference list you have a lot of old papers. You can try to identify if the information taken from the old papers are actual or not, and if there are not an update of the information.
Given the fact that the abstract and the conclusion part of the papers are the most read parts you can eliminate the unnecessary information and focus more on your findings also adding some recommendations for international readers which can learn from China context.
Overall, you have a good paper, it has good parts, and I consider that you just need to work a little bit to better present your research.
Good luck in your future research!
